# Rehabilitation after Hip Fracture Surgery: A Survey on Italian Physiotherapists’ Knowledge and Adherence to Evidence-Based Practice

**DOI:** 10.3390/healthcare11060799

**Published:** 2023-03-08

**Authors:** Fabio Santacaterina, Sandra Miccinilli, Silvia Sterzi, Federica Bressi, Marco Bravi

**Affiliations:** 1Research Unit of Physical and Rehabilitation Medicine, Department of Medicine and Surgery, Università Campus Bio-Medico di Roma, Via Alvaro del Portillo, 21, 00128 Rome, Italy; 2Department of Physical and Rehabilitation Medicine, Fondazione Policlinico Universitario Campus Bio-Medico, Via Alvaro del Portillo, 200, 00128 Rome, Italy

**Keywords:** rehabilitation, hip, fracture, physiotherapy, survey, exercise, management, multidisciplinary, orthogeriatric, elderly, falls

## Abstract

The average life expectancy of the Italian population has increased over the last decades, with a consequent increase in the demand for healthcare. Rehabilitation after hip fracture surgery is essential for autonomy, recovery, and reintegration into the social context. The aim of this study was to determine the level of knowledge and adherence to the recent treatment recommendations of the Italian physiotherapists. A web-based survey, composed of 21 items, was conducted and the frequencies and percentages of the responses were analyzed to evaluate if there was an integration and adherence to the recommendations of greater than 70%, with respect to the desired response. A total of 392 responses were collected and analyzed. Recommendations regarding the multidisciplinary approach, early mobilization, and progressive muscle strength training, achieved the desired value in the inpatient setting. Intensive rehabilitation and full weight bearing did not reach the threshold values. The results of this survey show a partial integration of the recommendations for rehabilitation after hip fracture surgery by Italian physiotherapists. Adherence seems to be better in the inpatient setting and with physiotherapists with higher levels of education.

## 1. Introduction

The demographic trend in the most developed countries shows an increase in life expectancy over the last 50 years [1].

Italy is among the countries with the highest life expectancy in the world. Between 2015 and 2020, Italian life expectancy was 83 years compared to the European average of 77 years [2]. For this reason, the Italian population is exposed to a greater risk of typical old-age-incurred pathologies, such as osteoporosis and its complications. Falls are very common in the elderly population [3] and can lead to injuries and fractures [4], even due to low-energy trauma [5]. Hip fractures are among the most fearsome fractures due to the related high mortality and disability and their impact on patients’ quality of life and public assistance costs [6].

In particular, Italian health care hip fracture related costs are higher and grow faster than myocardial infarction [7]. Furthermore, recent studies have shown that the number of hospitalizations for hip fractures is constantly increasing in Italy, especially in the population over 75 years of age [8].

The choice treatment after hip fracture is surgery, whereas the conservative approach is limited to patients with unstable clinical conditions, affected by terminal pathologies and to non-ambulatory patients [9]. An early surgical approach (24–48 h) is associated with less perceived pain and a reduced hospitalization period [10]. Similarly, an early rehabilitation approach is associated with better postoperative recovery [11].

The rehabilitation process represents a fundamental element after surgery for the recovery of the activities of daily life (ADL) and social reintegration. In this regard, there are several guidelines for the management of hip fractures, the most used being the NICE guidelines, published first in 2011 and then updated over the years [12]. Over the years, the need to implement the NICE guidelines has emerged both with regard to some aspects of patient management, and in consideration of the worldwide healthcare systems heterogeneity [13,14].

As for the Italian context, different guidelines were followed according to regional indications over the years. Starting from 2021, there was an increase in the production of recommendations for the management of patients with hip fractures, both nationally [15,16] and internationally [11,17]. These recommendations support a multidisciplinary approach in an orthogeriatric model for the rehabilitation of the patients with hip fractures. Moreover, an early mobilization is suggested with the promotion of a complete weight bearing (when possible). Furthermore, an intensive rehabilitation and a progressive muscle strength training proved to be effective [18].

In recent years, several surveys have been conducted to analyze the knowledge and skills of Italian physiotherapists in the musculoskeletal field [19,20,21,22].

To the best of our knowledge, no studies investigated whether these recommendations are actuated in clinical practice by Italian physiotherapists. Therefore, the aim of the present study was to explore the current approach of physiotherapists in the management of post-operative patients after hip fracture surgery (until 72 h) in Italy. The secondary aim of the present study was to analyze variables related to the surgical intervention, such as surgical type, weight-bearing concession, and who grants weight-bearing prescription.

The hypothesis was to find out a variable adherence to guidelines according to work setting, expertise level, and regional origin. The hypothesized variability is based on the heterogeneity of the regional care pathways present in Italy.

## 2. Materials and Methods

### 2.1. Study Design

A cross-sectional observational study was conducted according to the Strengthening the Reporting of Observational Studies in Epidemiology (STROBE) recommendations for reporting observational studies [23]. A web-based survey (Forms, Microsoft), made up of 21 items, addressed to Italian physiotherapists, was developed with the aim of investigating the rehabilitation approach in the post-surgical phase (within 72 h after surgery) of patients with hip fracture (ICD-11 NC72.Z) [24]. The project was registered on the Open Science Framework (OSF) website in October 2022. The study obtained the approval of the ethics committee of the Campus Bio-Medico University of Rome (Prot. PAR 73.22 OSS) on 25 October 2022 [23].

### 2.2. Questionnaire

A questionnaire composed of 21 items (Table 1) was based on the analysis of the recent guidelines and recommendations provided by SIOT (Società Italiana di Ortopedia e Traumatologia) [16], inter-society consensus promoted by SIGG (Società Italiana di Gerontologia e Geriatria) [15] and according to the most recent clinical practice guidelines [11,17]. The analysis of considered guidelines, showed some common considerations, such as: importance of a multidisciplinary approach for evaluation and treatment, full weight bearing promotion (where allowed), early mobilization (within the first 48 h), intensive rehabilitation approach (at least one session a day) and progressive muscle strength training (Table 2). The questionnaire was therefore set up to investigate how much Italian physiotherapists had included the above elements in their knowledge and clinical practice.

According to the International Handbook of Survey Methodology [25], for the face validity, the questionnaire was preliminarily submitted to six expert physiotherapists during acute phase management of patients with hip fractures, to three physiatrists, three orthopedic surgeons, and to three expert nurses in the field to request any needed modification or implementation. They worked independently and agreed with the final version of the questionnaire. Furthermore, for a stronger content validity of the questions, the final version of the questionnaire was delivered to 20 Italian physiotherapists for any feedback about structure and contents.

Questionnaires were carried out anonymously. The final version of the questionnaire was administered in the period from 1 November 2022 to 31 January 2023. The Italian version, used for this survey, is also available in the Appendix A.

### 2.3. Participants

All graduate physiotherapists declaring that they practiced the profession of physiotherapist in Italy were invited to participate in this survey. All subjects included in the analysis declared that they had read the informed consent and authorized the processing of data for scientific research purposes. Participating physiotherapists had the opportunity to leave their e-mail address to receive aggregated data once the questionnaire was closed. The mailing list of the technical-scientific association “Associazione Italiana di Fisioterapia” (AIFI) was used to disseminate the questionnaire.

### 2.4. Study Size

The study size was calculated based on recent surveys that analyzed the same study population (Italian physiotherapists) [22,26]. The formula proposed by Taherdoost [27] for the surveys sample size was therefore used. Considering the Italian physiotherapists population, it stands at 69,848 according to the FNOFI (Federazione Nazionale Ordini Fisioterapisti) [28]. The minimum answer number is 384 considering a 5% margin of error and a 95% confidence level.

### 2.5. Variables

The main objective of the study was to investigate the level of knowledge and implementation of the recommendations in the clinical practice of Italian physiotherapists. For this reason, in absence of a standardized threshold, the items that reached ≥70% of the desired response were considered implemented [22,29]. In item 12, the desired answer was “full weight bearing”. In Items 14, 15, and 16 the desired answer was “very useful”. In Item 17 the desired answer was “at least one session a day”, in Item 19 “Within 24 h” and “between 24 and 48 h”, in item 20 “progressive muscle strength training”, in item 9 and 21 the answer “yes”. The sub-analyses, carried out regarding the regional origin, the setting, and the level of training, aimed to understand how the recommendations are absorbed in everyday clinical practice [30] in order to guarantee an evidence-based approach throughout the Italian territory.

### 2.6. Analysis

The questionnaire did not include open-ended questions; therefore, all the results were analyzed using descriptive statistics to provide data on the frequency (counts, percentages) of the related answers. Microsoft Excel 365 v.2211 (Microsoft Corporation, Redmond, WA, USA) and SPSS Statistics for Windows, Version 26.0 (IBM Corp Armonk, NY, USA) were used for the analysis.

In addition to the global evaluation of the data, it was decided to carry out sub-analyses by filtering the responses based on the origin of the physiotherapists (Northern, Central, Southern Italy), based on the work setting (inpatient, outpatient and home-based therapies) and based on the level of expertise (starter or advanced).

## 3. Results

### 3.1. Demographics and Expertise

A total of 392 responses were analyzed. A total of 224 female (57%) and 168 male (43%) physiotherapists replied to the questionnaire. The mean age of responders was 38.2 ± 12.2 years. The average number of years of activity as a physiotherapist was 13.9 ± 5.8 years. Most physiotherapists (*n* = 344; 88%) declared that they attended continuing medicine education program (ECM) courses as postgraduate training, while 59 (15%) hold a Master of Science (MSc) degree; 157 physiotherapists (40%) attended a first level master’s degree, 7 (2%) the second level master’s degree, and 2 physiotherapists (1%) declared to be in possession of a Doctor of Philosophy (PhD) title. Regarding the Orthopaedic Manipulative Physical Therapist (OMPT) title, issued by International Federation of Orthopaedic Manipulative Physical Therapists Incorporated (IFOMT), 56 physiotherapists (14%) held the title while 39 (10%) were attending the enabling first level master course. In our sub-analysis we divided responders into two groups: 201 (51%) starters (bachelor’s degree and post graduate courses) and 191 (49%) advanced (Master of Science, Doctor of Philosophy, master’s degree).

### 3.2. Geographical Spread

Answers were received from all of the 20 Italian regions. As for the demographic disposition of the respondents on the territory, there is a greater frequency of responses from Lombardia (*n* = 77; 20%) and Lazio (*n* = 88; 22%) as shown in Figure 1. Regarding the sub-analysis of the data based on the origin of the questionnaires, the following division was made:Northern Italy (*n* = 204): Valle D’Aosta (2), Piemonte (33), Lombardia (77), Trentino Alto-Adige (8), Friuli-Venezia Giulia (16), Veneto (29), Liguria (13), Emilia-Romagna (26);Central Italy (*n* = 119): Toscana (19), Marche (3), Umbria (4), Lazio (88), Abruzzo (5);Southern Italy (*n* = 69): Campania (12), Basilicata (3), Molise (3), Calabria (4), Puglia (15), Sicilia (22), Sardegna (10);

### 3.3. Setting

According to work setting, participants could be divided into three main categories: hospital and clinics (inpatient *n* = 197, 50%), outpatient (*n* = 141, 36%), and home-based service (*n* = 48, 14%).

Regarding hospital and clinics, answers came largely from physiotherapists employed in public hospitals (*n* = 80; 20%), while 60 (16%) answers came from physiotherapist from a private hospital, 17 (4%) from a university hospital, and 40 (10%) from a private clinic.

With reference to outpatient setting, 33 physiotherapists (8%) were employed in a public outpatient clinic, 108 (33%) in a private outpatient clinic, and regarding home-based services, 32 (8%) physiotherapists declared delivering public home-based treatments and 22 (6%) private home-based treatments.

### 3.4. Questionnaire Results

Concerning the monthly frequency treatment of patients with hip fractures (Item 8), it emerged that, stratifying the data by setting, 61% of the inpatient group, 40% of the home-based and only 20% of the outpatient group, stated that they treated patients with hip fractures between often and very often.

#### 3.4.1. Multidisciplinary Team

From the item 9 analysis it emerged that 58% (*n* = 228) declared that they work in a multidisciplinary team that carried out the evaluation, while 42% (*n* = 164) declared that no multidisciplinary team was present in their work setting. Considering only the inpatient setting sub-analysis, 81% of the interviewees (*n* = 159) reported the presence of the multidisciplinary team, while 19% (*n* = 38) did not work in a multidisciplinary team. Stratifying the inpatient group by geographical area, it emerged that 87% (*n* = 87) of Northern inpatients physiotherapists, 83% (*n* = 48) of Central, and 83% (*n* = 24) of Southern worked in a multidisciplinary team (Figure 2).

Regarding item 14, which analyzed how much the multidisciplinary team positively influences rehabilitation, 69% of total sample chose “very useful”, this value increasing to 70% (*n* = 138) if we consider the inpatient subgroup and to 72% (*n* = 101) if we consider the outpatient group, and to 59% (*n* = 32) in the home-based group. In the expertise level sub-analysis, 68% (*n* = 136) of starter physiotherapists and 71% (*n* = 135) of advanced physiotherapists considered “very useful” the multidisciplinary approach (Table 3).

#### 3.4.2. Weight Bearing

The most frequent surgery (item 11) in patients after hip fracture was hip replacement 66% (*n* = 258), while nailing reached 34% (*n* = 134). With reference to the inpatient domain, there was a change in values with hip replacement equal to 59% (*n* = 116), and with nailing equal to 41% (*n* = 81).

Partial weight bearing is usually the most prescribed (item 12) (55%, *n* = 214), followed by the full indication (35%, *n* = 137) and by prohibited weight bearing (9%, *n* = 34). Analyzing the data according to the type of intervention, it was observed that in the case of hip arthroplasty there is a protected prescription in 57% (*n* = 146), full weight bearing in 39% (*n* = 100), and prohibition in 3% (*n* = 7). In the event of nailing, the most frequent type is in any case the protected 51% (*n* = 68), full in 28% (*n* = 37), and prohibited in 10% (*n* = 27).

The weight-bearing prescription (item 13) was carried out in 81% (*n* = 316) of the answers analyzed by the orthopedic surgeon, in 16% (*n* = 64) by the physiatrist, and in 1% (*n* = 3) by the geriatrician.

#### 3.4.3. Rehabilitation Session

The optimal number of daily sessions in the acute phase (item 17) was found to be at least one session a day (45%, *n* = 175), followed by one session a day (39%, *n* = 154), and by three sessions a week (15%, *n* = 59). The data relating to this domain undergo variations in the starter, outpatient, and home-based subgroups (Table 4).

The duration of a physiotherapy session (item 18) was found to be between 30 and 45 min of treatment (34%, *n* = 133). Table 5 shows frequency variations according to the sub-analyses. In the inpatient group the optimal duration was between 15 and 30 min (45%, *n* = 89).

#### 3.4.4. Therapeutic Features

Early mobilization within 48 h post-surgery (item 15) was more useful than late mobilization in 70% (*n* = 274) of the total questionnaires analyzed. This data turns out to be the most frequent in all the sub-analyses included, reaching the highest value (74% *n* = 138) in the inpatient subset (Table 6).

The intensive rehabilitation approach (Item 16) is considered very useful by 44% (*n* = 171) of the interviewees, while 45% (*n* = 1178) believe that it is quite useful when compared with a non-intensive approach. The only subgroups in which there is a higher frequency in the “very useful” response concern physiotherapists from central and southern Italy, working in an inpatient setting or with an advanced expertise (Table 7).

Item 19 investigated the post-operative verticalization of the patient operated for hip fracture. The most frequent period was between 24 and 48 h: 61% (*n* = 239). This remains largely the most frequent in all the analyzed subgroups (Table 8).

The most frequently used exercise modality (item 20) was progressive muscle strength training 72% (*n* = 283). Weight-bearing exercises were chosen by 48% (*n* = 198), while 44% (*n* = 172) used gait training exercises. In this question, the physiotherapists had the chance to answer “other” (18%, *n* = 68), specifying the answer. Many answers included isometric quadriceps contractions, manual therapy for ROM recovery, lymphatic drainage, balance, and proprioceptive exercises, and only one indicated muscle stretching.

The intervention of other healthcare professionals to promote patient mobilization in the acute phase was considered an important element (55%, *n* = 214). A total of 65% (*n* = 133) of the Northern group, 46% (*n* = 55) of the Central, and 38% (*n* = 26) of the Southern group, also thought it was important. Analyzing the subgroups related to the work setting, 65% (*n* = 128) of the inpatient group believed that the intervention of other professions is important, the home-based group 50% (*n* = 27), while in the outpatient groups 42% (*n* = 59). With respect to the expertise subgroups, the 52% (*n* = 99) of advanced physiotherapists and the 57% (*n* = 115) of starters considered the intervention of other professional figures important.

## 4. Discussion

In 1996, evidence-based practice (EBP) was defined as the therapeutic action that takes into consideration the best evidence in the literature, the clinician’s expertise, and the patient’s values [31]. However, the implementation of the EBP remains a great challenge in physiotherapy as it is connected to health pathways, the complexity of the physiotherapy subject, and access to continuing education programs [32]. In this study, we tried to analyze how the recommendations for hip fractured patients are integrated into the knowledge and clinical practice of Italian physiotherapists during rehabilitation in the acute phase after hip fracture surgery.

The multidisciplinary team, that carries out the multidimensional assessment is present especially in the inpatient setting (81%). This result is comforting, since a multidisciplinary approach in the hospital phase has a proven effectiveness in the reduction in “poor outcomes” (e.g., death or deterioration in residential status) if compared with the usual care [33]. Furthermore, the majority of physiotherapists consider the multidisciplinary approach very useful (Item 14), confirming that this aspect is not only integrated into the clinical practice, but its importance is perceived. Nevertheless, a critical fact that emerged from this survey is the scarce presence of the geriatrician specialist within the multidisciplinary team. A recent study, in fact, shows how a multidisciplinary team in an orthogeriatric model contributes to the reduction in multi-effect drugs and perioperative complications [34]. The role of the geriatrician is therefore essential in the management of the patient with hip fracture, also because patients have many other concomitant pathologies in addition to fracture.

Weight-bearing prescription is almost carried out by Orthopedics and the most frequent indication is partial weight bearing in case of hip arthroplasty or nailing. This is in contrast with the recent recommendations that promote a full weight-bearing prescription [11,16,17]. Several studies demonstrate how full weight-bearing concession shortened length of stay in hospital, increased walking abilities [35] and, in contrast, weight-bearing restrictions induce a loss of mobility and should be avoided [36]. It is equally true that integrating evidence into clinical practice can take many years [37] and just as many years to stop using non-evidence-based approaches [38].

Less than half of the respondents considered intensive rehabilitation very useful. However, the response rate increased among physiotherapists working in the inpatient setting and those who underwent advanced training.

Certainly, this is an aspect that has not yet been fully embraced, despite evidence of a reduction in length of stay and an increased probability with intensive rehabilitation [12]. The partial absorption of intensive rehabilitation is also confirmed by responses regarding the optimal daily session dose: in fact, less than half answered “at least one per day”, whereas this is the optimal intensity according to recent recommendations [16].

Another interesting aspect is the length of sessions, which should last between 30 and 45 min. However, if we consider the responses from the inpatient and advanced groups, the duration is usually between 15 and 30 min. A possible explanation is that these groups provide at least one treatment per day, and therefore reduce the average time of a session to avoid overloading the patient.

Early mobilization is widely considered very helpful by most physiotherapists in accordance with evidence [39]. This element is confirmed by the timing of verticalization; in fact, most physiotherapists stated that they stand up the patient within 48 h after surgery. Being able to mobilize the patient as soon as possible to avoid prolonged bed rest is also a useful recommendation for preventing bedsores [17].

The most proposed exercise by the interviewed physiotherapists is certainly progressive muscle strength training (72%). Progressive high-intensity structured exercise for the recovery of muscle strength is an essential element and is strongly recommended [17,18].

In general, the interventions focused on the recovery of mobility, especially in the acute phase, showed their effectiveness in improving walking speed, also through gait training [40] which was used by only 44% of the interviewees. Probably the reason why less than half of physiotherapists chose gait training is related to the still low prescription of full weight-bearing.

The usefulness of involving other healthcare professionals to promote patient mobilization is, in our opinion, an essential element to ensure the best recovery after hip fracture surgery. This vision is shared by 55% of the interviewees but certainly requires more attention as it is also a crucial aspect of the multidisciplinary approach [15].

Data collected showed a good integration of the recommendations about the early mobilization, multidisciplinary approach, and use of structured exercise. Some aspects, such as full weight bearing, intensive rehabilitation approach, implementation of gait training, and involvement of other health professionals in the mobilization, need greater integration in the knowledge and in the clinical practice of Italian physiotherapists.

We also need to consider barriers in EBP implementation such as economic aspects, insufficient skills, and knowledge [41]. At the same time, work overload seems to be more impactful [42]. In 2021, a systematic review by Paci et al. [43], highlighted the need for universities to fill gaps in university programs about statistics, research, and scientific English [37]. It is our opinion that degree courses in Italy should implement these teachings in order to understand more in depth the importance of an EBP approach in clinical practice.

### 4.1. Study Implications

This study showed that the recommendations in the clinical practice of Italian physiotherapists have not yet been completely absorbed. It is often assumed that if an element is recommended then it is also implemented in the clinic, Italian physiotherapists should consider including elements such as the intensive approach and exercises aimed at early recovery of ambulation. Furthermore, physiotherapists should consider the importance of involving other professional figures in the patient’s mobilization as it could contribute to early and continuous mobilization even in moments outside the physiotherapy intervention. In this regard, this work offers a reflection for the entire multidisciplinary team that takes charge of the patient with hip fracture outcomes; full weight bearing is often not allowed and the reason behind this choice should be explored, in order to ascertain whether it is linked to the patient’s clinical conditions or to an incomplete adherence to the recommendations. Another relevant clinical element is the scarce presence of the geriatrician in the multidisciplinary team: it is our opinion that the lack of such a cross-sectional figure does not contribute to a speedy recovery and complete management of the patient.

Furthermore, the survey conducted shows that there are differences in adherence to the recommendations based on the level of training of physiotherapists: the university course must increasingly tend to develop knowledge and skills in the field of scientific research, scientific English, and the application of good clinical practice. It could be interesting to submit the questionnaire to other countries to assess whether the level of adherence is similar or differs from the Italian one in order to plan interventions aimed at improving the knowledge and skills of Italian and worldwide physiotherapists.

The implementation of the recommendations can have a direct effect on the improvement of the treatments offered, thus improving the patient’s outcomes in terms of recovery of function and reintegration into the social context of belonging.

### 4.2. Limitations

Some study limitations, such as the lack of analysis of pain management in the acute phase, which represents an essential element for patient management, are present in this study. Furthermore, the evaluation aspect of the clinical scales has not been investigated, such as, for example, the use of the Cumulative Ambulation Score (CAS) [44] and the management of situations in which there are contraindications to the patient’s mobilization for clinical reasons. Future works should implement these aspects to offer a complete vision from a physiotherapy point of view. It will also be appropriate to carry out a survey that analyzes the post-acute rehabilitation path of the patient with surgery outcomes for hip fracture where secondary prevention of falls and balance training play an important role and analyze the role of other important rehabilitation figures who contribute in the post-acute phase, such as occupational therapists.

## 5. Conclusions

This study seems to demonstrate a partial integration of the recommendations regarding the multidisciplinary team, early mobilization, and the use of exercises for progressive strength recovery by Italian physiotherapists. The intensive rehabilitation approach and full weight bearing require greater integration. Sub-analyses seem to suggest that adherence is higher among physiotherapists working in a hospital setting and among those with higher levels of education.

A greater presence of the geriatrician in the multidisciplinary team and the involvement of other healthcare figures for mobilization is desirable.

## Figures and Tables

**Figure 1 healthcare-11-00799-f001:**
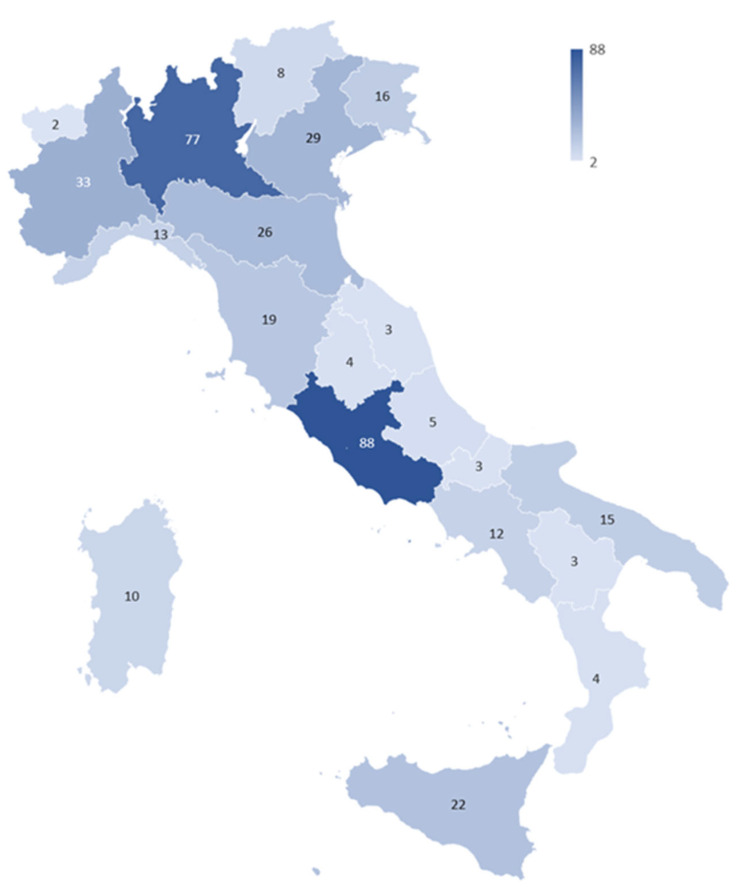
Graphical representation of the survey participants distribution among Italian regions. The regions with the darkest color represent those with the greatest number of participants.

**Figure 2 healthcare-11-00799-f002:**
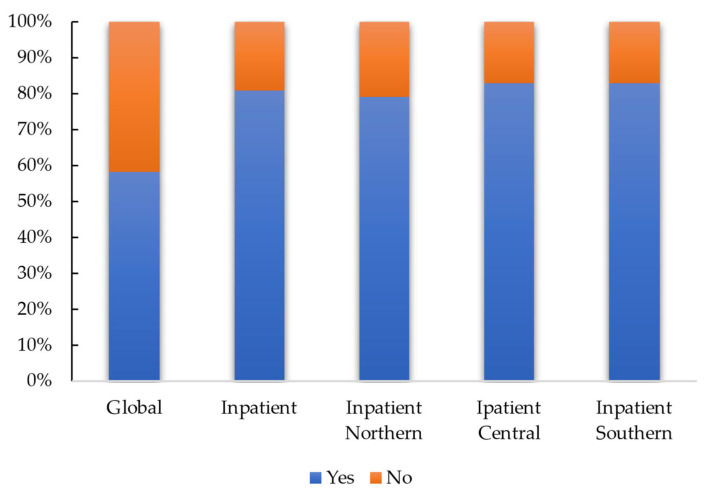
The graph shows the percentage of responses regarding the presence of the multidisciplinary team at national level (global) and focusing the analysis on physiotherapists who work in a hospital setting (inpatient), among these stratifying the data geographically. Distribution: as regards the composition of the multidisciplinary team (item 10), it emerged that the most represented figure is the physiotherapist, and the least represented is the geriatrician. The distribution of composition is shown in Figure 3 (more than one possible answer was optionable).

**Figure 3 healthcare-11-00799-f003:**
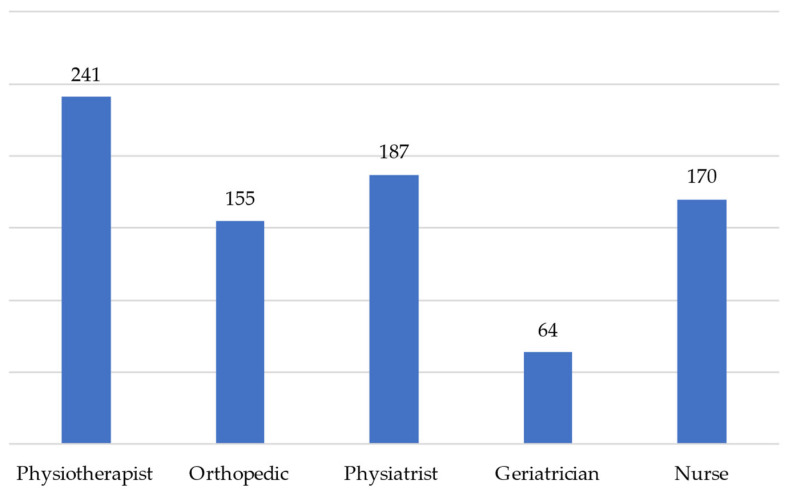
Absolute frequency of answers relating to the composition of the multidisciplinary team.

**Table 1 healthcare-11-00799-t001:** Questionnaire used in the survey (English translation).

Questions	Answers
1. In which region do you work?	
2. Gender	Male
	Female
3. How old are you?	
4. How many years have you been working as a physiotherapist?	
5. What was your postgraduate education?	Post graduate courses
	Master of Science
	Master’s degree
	PhD
6. Are you in possession of the OMPT title?	Yes No Masters’ student
7. In which setting does your clinical activity mainly take place?	Public Hospital Private Hospital University Hospital Private clinic Public outpatient service Private outpatient service Public-home-based service Private-home-based service
8. How often do you treat patients with hip fracture?	Very often (more than 10 patients per month) Often (between 5 and 10 patients per month) Occasionally (between 1 and 5 patients per month) Rarely (between none and 1 patient per month)
9. Is there a multidisciplinary team in the place where you practice that carries out the evaluation of the patient with a hip fracture?	Yes No
10. If yes at question 9. who is the multidisciplinary team composed of?	Physiotherapist Orthopedic MD Physiatrist MD Geriatrician MD Nurse Other (specify)
11. What type of surgery is most common in the facility where you work?	Nailing Hip Arthroplasty
12. Which type of weight bearing is most frequent after the surgery you selected in the previous question?	Non-weight-bearing Partial weight-bearing Full weight bearing No indication
13. What specialist give weight bearing indication for the patient?	Orthopedic MD Physiatrist MD Geriatrician MD Other figure (please indicate which)
14. How much do you think the multidisciplinary team can positively affect patient’s recovery?	Very Enough Little Useless
15. How much do you think that an early mobilization program (within 48 h) compared to a late one (over 48 h) can positively affect the recovery of the patient with a hip fracture?	Very Enough Little Useless
16. How much do you think that an intensive rehabilitation program compared to a non-intensive one can positively affect the recovery of the patient with a hip fracture?	Very Enough Little Useless
17. In your opinion, what is the optimal number of physiotherapy sessions in the acute phase for the patient with a hip fracture?	At least one session a day One session a day Three sessions a week Two sessions a week One session a week
18. How long does an average physiotherapy session last for this type of patient in the acute phase (48–72 h) in the place where you practice?	0–15 min 15–30 min 30–45 min 45 min–1 h Over 1 h
19. How long does it take for a patient to be verticalized after surgery for a hip fracture?	Within 24 h Between 24 and 48 h Between 48 and 72 h Over 72 h
20. What type of exercise do you most frequently propose to these patients in the first 48–72 h? (Insert more answers if appropriate)	Progressive muscle strength training Weight-bearing exercises Gait training Other (specify)
21. Is the intervention of other professional figures important for the mobilization of the patient with a hip fracture in the acute phase?	Yes (specify) No

**Table 2 healthcare-11-00799-t002:** Knowledge and clinical practice items of evidence-based recommendations.

Recommendations	Knowledge	Clinical Practice
Multidisciplinary approach	ITEM 14	ITEM 9
Full weight-bearing	-	ITEM 12
Early mobilization	ITEM 15	ITEM 19
Intensive rehabilitation	ITEM 16	ITEM 17
Progressive strength training	-	ITEM 20

**Table 3 healthcare-11-00799-t003:** Answers referred to item 14: How much do you think the multidisciplinary team can positively affect the patient’s recovery?

ITEM 14	Answers			
	Very	Enough	Little	Useless
Global	69% (271)	28% (108)	3% (12)	0% (1)
Work setting				
Inpatient	70% * (138)	28% (55)	1% (3)	1% (1)
Outpatient	72% * (101)	23% (32)	6% (8)	0% (0)
Home-based	59% (32)	39% (21)	2% (1)	0% (0)
Expertise				
Starter	68% (136)	29% (58)	3% (6)	0% (1)
Advanced	71% * (135)	26% (50)	3% (6)	0% (0)

* Percentages that exceed the threshold value.

**Table 4 healthcare-11-00799-t004:** Answers referred to item 17: In your opinion, what is the optimal number of physiotherapy sessions in the acute phase for a patient with hip fracture?

ITEM 17		Answers			
	at Least 1 a Day	1 a Day	3 a Week	2 a Week	1 a Week
Global	45% (175)	39% (154)	15% (59)	1% (3)	0% (1)
Geographical spread					
Northern	44% (90)	42% (85)	14% (28)	0% (1)	0% (0)
Central	43% (51)	39% (46)	17% (20)	1% (1)	1% (1)
Southern	49% (34)	33% (23)	16% (11)	1% (1)	0% (0)
Work setting					
Inpatient	54% (106)	38% (75)	8% (15)	1% (1)	0% (0)
Outpatient	37% (52)	40% (57)	22% (31)	1% (1)	0% (0)
Home-based	31% (17)	41% (22)	24% (13)	2% (2)	2% (1)
Expertise					
Starter	43% (87)	43% (86)	13% (27)	0% (1)	0% (0)
Advanced	46% (88)	36% (68)	17% (32)	1% (2)	1% (1)

**Table 5 healthcare-11-00799-t005:** Answers referred to item 18: How long does an average physiotherapy session last for this type of patient in the acute phase (48–72 h) in the place where you practice?

ITEM 18		Answers			
	0–15 min	15–30 min	30–45 min	45–1 h	Over 1 h
Global	5% (20)	30% (119)	34% (133)	26% (102)	5% (18)
Geographical spread					
Northern	5% (10)	35% (72)	36% (74)	21% (43)	2% (5)
Central	5% (6)	27% (32)	25% (30)	33% (39)	10% (12)
Southern	6% (4)	22% (15)	42% (29)	29% (20)	1% (1)
Work setting					
Inpatient	5% (10)	45% (89)	22% (44)	20% (40)	7% (14)
Outpatient	3% (4)	16% (22)	43% (60)	36% (51)	3% (4)
Home-based	11% (6)	15% (8)	54% (29)	20% (11)	0% (0)
Expertise					
Starter	5% (10)	29% (59)	40% (81)	21% (42)	4% (9)
Advanced	5% (10)	31% (60)	27% (52)	31% (60)	5% (9)

**Table 6 healthcare-11-00799-t006:** Answers referred to item 15: How much do you think that an early mobilization program (within 48 h), compared to a late one (over 48 h), can positively affect the recovery of the patient with a hip fracture?

ITEM 15	Answers			
	Very	Enough	Little	Useless
Global	70% * (274)	25% (99)	4% (16)	1% (3)
Geographical spread				
Northern	69% (140)	27% (55)	3% (7)	1% (2)
Central	73% * (87)	22% (26)	4% (5)	1% (1)
Southern	68% (47)	26% (18)	6% (4)	0% (0)
Work setting				
Inpatient	74% * (146)	22% (43)	4% (7)	1% (1)
Outpatient	66% (93)	27% (38)	6% (9)	1% (1)
Home-based	65% (35)	33% (18)	0% (0)	2% (1)
Expertise				
Starter	68% (137)	28% (56)	4% (8)	0% (0)
Advanced	72% * (137)	23% (43)	4% (8)	2% (3)

* Percentages that exceed the threshold value.

**Table 7 healthcare-11-00799-t007:** Answers referred to item 16: How much do you think that an intensive rehabilitation program compared to a non-intensive one can positively affect the recovery of the patient with a hip fracture?

ITEM 16	Answers			
	Very	Enough	Little	Useless
Global	44% (171)	45% (178)	9% (35)	2% (8)
Geographical spread				
Northern	42% (86)	46% (94)	10% (21)	1% (3)
Central	45% (53)	43% (51)	9% (11)	3% (4)
Southern	46% (32)	48% (33)	4% (3)	1% (1)
Work setting				
Inpatient	46% (90)	44% (86)	8% (16)	3% (5)
Outpatient	45% (64)	43% (61)	11% (15)	1% (1)
Home-based	31% (17)	57% (31)	7% (4)	4% (2)
Expertise				
Starter	39% (78)	46% (92)	13% (26)	2% (5)
Advanced	49% (93)	45% (86)	5% (9)	2% (3)

**Table 8 healthcare-11-00799-t008:** Answers referred to item 19: How long does it take for a patient to be verticalized after surgery for a hip fracture?

ITEM 19	Answers			
	Within 24 h	24–48 h	48–72 h	Over 72 h
Global	18% (69)	61% (239)	16% (64)	5% (20)
Geographical spread				
Northern	21% (43)	55% (113)	19% (38)	5% (10)
Central	13% (15)	69% (82)	13% (16)	5% (6)
Southern	16% (11)	64% (44)	14% (10)	6% (4)
Work setting				
Inpatient	14% (28)	62% (122)	18% (35)	6% (12)
Outpatient	23% (32)	62% (88)	12% (17)	3% (4)
Home-based	17% (9)	54% (29)	22% (12)	7% (4)
Expertise				
Starter	17% (35)	61% (122)	16% (32)	6% (12)
Advanced	18% (34)	61% (117)	17% (32)	4% (8)

## Data Availability

The data will be made available upon reasonable request to the corresponding author.

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
