# Peer review of "Rehabilitation after Hip Fracture Surgery: A Survey on Italian Physiotherapists’ Knowledge and Adherence to Evidence-Based Practice"

_healthcare, 2023, doi:10.3390/healthcare11060799_

Round 1
Reviewer 1 Report
Rehabilitation after hip fracture surgery: a survey on Italian physiotherapists’ knowledge and adherence to evidence-based practice
GENERAL COMMENTS
Thank you for allowing me to review this manuscript. The manuscript adhere the Healthcare Data Policy. The aim of this study was to investigate the rehabilitation after hip fracture surgery and determine the knowledge and adherence to the recent treatment recommendations of the Italian physiotherapists.
A cross-sectional observational study was conducted throughout the creation of a web-based survey, made up of 21 items, addressed to Italian physiotherapists, with the aim of investigating the rehabilitation approach in the post-surgical phase (within 72 hours after surgery) of patients with hip fracture.
This is an interesting research topic with potential utilization across disciplines and relevant to the journal. In my opinion, the paper would need minor changes. Revisions will be necessary. Improve the organization of your paper using the following guidelines.
INTRODUCTION
- Abstract: - What is the current state of evidence on the therapeutic approach for the condition studied in the study? What's new in the scientific literature with this manuscript? Include in introduction. How well is the paper integrated with current research.
-The introduction needs to better indicate the existing evidence on therapeutic approaches for the condition studied. The use of systematic reviews is recommended.
-The manuscript not include a hypothesis: HOW? WHY? Include. Explain the hypothesis.
METHODS
- Who has diagnosed the condition and on what basis or criteria he has been diagnosed?
-Include the ICD classification of the disease.
DISCUSSION
-Include the strengths of the study.
-Thus, the reader can decide whether the results have relevance and applicability in their daily practice. Question? ARE THE RESULTS APPLICABLE IN PRACTICE? Add. Include the implications/applications of the study for the field movement sciences.
- Include the clinical significance of this study over clinicians, patients, and researchers after the study hypothesis.
Reviewer 2 Report
Although they do not allow a greater degree of depth to be reached, the survey on the profession's usual actions is an important point to know the treatment preferences in the field.
The introduction is good in general terms, although some references are missing (I suggest line 65), or citing other similar initiatives on which the authors have based the development of their questionnaire. I also recommend reference the validation of this questionnaire and, if it does not exist, an analysis of reliability and validity of the different questions and their response rates. Also, a previous based justification of different clinical ranged variables in the questionnaire would be appreciated.
It should not be forgotten that there may be biases determined by the preferences or possibilities of users and physical therapists. I recommend exploring this question further. In that way, some variables in the questionnaire could have been correlated to refine the relationship between factors such as the work environment and the type/preferences of physiotherapy (time, modality, etc.).
I recommend the use of STROBE statement for reporting descriptive results.
Occupational therapy is not cited within the professionals in charge of patient recovery, and OT is especially relevant in elderly patients, although I understand it is because of the early post-intervention phase.
The figure captions should be self-explanatory. They are excessively brief.
